# A Multi-Country, Single-Blinded, Phase 2 Study to Evaluate a Point-of-Need System for Rapid Detection of Leishmaniasis and Its Implementation in Endemic Settings

**DOI:** 10.3390/microorganisms9030588

**Published:** 2021-03-12

**Authors:** Prakash Ghosh, Abhijit Sharma, Narayan Raj Bhattarai, Kumar Abhishek, Thilini Nisansala, Amresh Kumar, Susanne Böhlken-Fascher, Rajashree Chowdhury, Md Anik Ashfaq Khan, Khaledul Faisal, Faria Hossain, Md. Rasel Uddin, Md. Utba Rashid, Shomik Maruf, Keshav Rai, Monica Sooriyaarachchi, Withanage Lakma Kumari Abhayarathna, Prahlad Karki, Shiril Kumar, Shalindra Ranasinghe, Basudha Khanal, Satyabrata Routray, Pradeep Das, Dinesh Mondal, Ahmed Abd El Wahed

**Affiliations:** 1Nutrition and Clinical Service Division, International Centre for Diarrhoeal Disease Research, Dhaka 1212, Bangladesh; r.chowdhury@icddrb.orgb (R.C.); anik.ashfaq@gmail.com (M.A.A.K.); otivuj@gmail.com (K.F.); faria109@gmail.com (F.H.); rasel.uddin@icddrb.org (M.R.U.); utba.rashid777@outlook.com (M.U.R.); shomik_stj@yahoo.com (S.M.); 2PATH, 15th Floor, Dr. Gopal Das Bhawan, 28 Barakhamba Road, New Delhi 110001, India; asharma@path.org (A.S.); akumar@path.org (A.K.); sroutray@path.org (S.R.); 3B. P. Koirala Institute of Health Sciences, Dharan 56700, Nepal; bhattarai03@yahoo.com (N.R.B.); raikeshav@hotmail.com (K.R.); prahladkarki2000@yahoo.com (P.K.); basudhak@gmail.com (B.K.); 4Department of Molecular Biology, ICMR-Rajendra Memorial Research Institute of Medical Sciences, Agamkuan, Patna 800007, India; abhisinghbhu41@gmail.com (K.A.); drshiril2013@gmail.com (S.K.); drpradeep.das@gmail.com (P.D.); 5Department of Microbiology, University of Sri Jayewardenepura, Gangodawila, Nugegoda 10250, Sri Lanka; thilini.ns90@gmail.com; 6Department of Animal Science, Division of Microbiology and Animal Hygiene, Georg-August-University of Goettingen, D-37077 Goettingen, Germany; susanne.boehlken-fascher@agr.uni-goettingen.de; 7Faculty of Medicine, University of Leipzig, D-04103 Leipzig, Germany; 8Institute of Animal Hygiene and Veterinary Public Health, University of Leipzig, D-04103 Leipzig, Germany; 9Department of Parasitology, University of Sri Jayewardenepura, Gangodawila, Nugegoda 10250, Sri Lanka; soori_w@yahoo.com (M.S.); lakma.ab@gmail.com (W.L.K.A.); ishalindra@sjp.ac.lk (S.R.); 10Department of Microbiology, Indira Gandhi Institute of Medical Sciences, Sheikhpura, Patna 800014, India; 11Laboratory Sciences and Services Division, International Centre for Diarrhoeal Disease Research, Dhaka 1212, Bangladesh

**Keywords:** leishmaniasis, RPA assay, mobile suitcase laboratory, diagnosis, point-of-need

## Abstract

With the advancement of isothermal nucleic acid amplification techniques, detection of the pathogenic DNA in clinical samples at point-of-need is no longer a dream. The newly developed recombinase polymerase amplification (RPA) assay incorporated in a suitcase laboratory has shown promising diagnostic efficacy over real-time PCR in detection of *leishmania* DNA from clinical samples. For broader application of this point-of-need system, we undertook a current multi-country diagnostic evaluation study towards establishing this technique in different endemic settings which would be beneficial for the ongoing elimination programs for leishmaniasis. For this study purpose, clinical samples from confirmed visceral leishmaniasis (VL) and post-kala-azar dermal leishmaniasis (PKDL) patients were subjected to both real-time PCR and RPA assay in Bangladesh, India, and Nepal. Further skin samples from confirmed cutaneous leishmaniasis (CL) patients were also included from Sri Lanka. A total of 450 clinical samples from VL patients, 429 from PKDL patients, 47 from CL patients, and 322 from endemic healthy/healthy controls were under investigation to determine the diagnostic efficacy of RPA assay in comparison to real-time PCR. A comparative sensitivity of both methods was found where real-time PCR and RPA assay showed 96.86% (95% CI: 94.45–98.42) and 88.85% (95% CI: 85.08–91.96) sensitivity respectively in the diagnosis of VL cases. This new isothermal method also exhibited promising diagnostic sensitivity (93.50%) for PKDL cases, when a skin sample was used. Due to variation in the sequence of target amplicons, RPA assay showed comparatively lower sensitivity (55.32%) than that of real-time PCR in Sri Lanka for the diagnosis of CL cases. Except for India, the assay presented absolute specificity in the rest of the sites. Excellent concordance between the two molecular methods towards detection of *leishmania* DNA in clinical samples substantiates the application of RPA assay incorporated in a suitcase laboratory for point-of-need diagnosis of VL and PKDL in low resource endemic settings. However, further improvisation of the method is necessary for diagnosis of CL.

## 1. Introduction

With the auspices of holistic efforts followed by the London Declaration, strides are ongoing to alleviate the scourge driven by (Neglected Tropical Diseases) NTDs in poverty stricken endemic tropics. Among 17 parasite-borne NTDs recognized by the WHO, leishmaniasis ranks next to malaria as the second worst in the age-standardized disability-adjusted life years (DALYs), and requires control mechanisms and tools including drugs, vaccine, diagnostics and vector control agents as well as strategies towards absolute elimination [1,2]. This vector-borne disease is comprised of three major forms including visceral leishmaniasis or kala-azar (VL), cutaneous leishmaniasis (CL), and mucocutaneous leishmaniasis affecting disproportionately both the new and old worlds, where more than 20 leishmania species are associated with disease pathogenesis [3]. In the Indian subcontinent, *Leishmania donovani* (LD) is the major causative agent, which is responsible for both visceral leishmaniasis and cutaneous leishmaniasis [4]. Besides, another form of leishmaniasis associated with dermatological complication that usually develops following successful treatment through anti-leishmanials of VL patients is coined as post kala-azar dermal leishmaniasis (PKDL) [5]. In the Indian subcontinent, Bangladesh, India, and Nepal harbor around 30% of the total visceral leishmaniasis cases worldwide, whereas LD causes cutaneous leishmaniasis in Sri Lanka [6]. Interestingly, the microsatellite analysis found that LD isolates from Sri Lanka are clustered together and close to a group containing LD isolates from Bangladesh, India, and Nepal [7]. These three countries have been jointly working to eliminate VL through implementing the kala-azar elimination program since 2005 [8]. To date, substantial success has been achieved in VL case reduction through the holistic approach of the program [9]. Now, towards cementing the success of the program and curbing further transmission, surveillance in the area harboring active infection towards a disease-free area would be highly appreciated. Therefore, an accurate, rapid, and simple diagnostic tool is imperative, which can be implemented at decentralized facilities for mass surveillance, diagnosis, and to cure assessment as well, if possible.

To date, microscopic detection of LD bodies in spleen or lymph node aspirates, sometimes in combination with culture techniques, remains the gold standard for laboratory diagnosis of VL [10]. Likewise, the dermatological involvement both in PKDL and CL require visualization of LD bodies in skin lesions for confirmed diagnosis [11,12]. However, these highly invasive and sub-optimally sensitive methods are performed at tertiary healthcare facilities only. To overcome the pitfalls of microscopy-based methods, several serological methods such as DAT, IFAT, and rK39 RDT have been developed, however, these methods are indecisive towards diagnosis of both VL and PKDL without clinical assessment [10]. Moreover, no ancillary serological diagnostic test is available for CL, due to the low level of circulating antibodies [13].

With the advancement of molecular technologies, several PCR-based methods have been developed to detect LD DNA in clinical samples of VL, PKDL and CL patients [11,14,15,16,17]. Notwithstanding, the promising diagnostic performance of these methods, requirement of well-equipped facilities and trained personnel, and lack of standardized protocols, limit their application for routine diagnosis. To meet the inadequacies of available molecular methods, recently a rapid, accurate and field feasible isothermal assay was developed [18,19]. This new isothermal assay involving recombinase enzyme or recombinase polymerase amplification assay (RPA), is widely regarded as a promising alternative technology to real-time PCR for point-of-need use as it provides results faster, amplifies nucleic acid at constant lower temperature, and is accomplished with less expensive as well as simpler equipment [20]. Discerning the multifarious advantages of RPA assay, we exploited this assay for detection of LD DNA in clinical samples. Our developed LD RPA assay showed equivalent diagnostic performance to real-time PCR towards detection of LD DNA in blood and skin of VL and PKDL patients, respectively [21]. Recently, Gunaratna et al. also showed promising efficiency of this assay in detection of LD DNA in skin lesions of CL patients [22]. To enable this simple method for point-of-need deployment, the LD RPA assay has been incorporated in a mobile suitcase laboratory. Furthermore, we established this point-of-need system successfully at primary healthcare settings for real-time diagnosis of suspected VL patients [21].

The multifaceted advantages of RPA assay incorporated in the suitcase laboratory accentuated further application of this diagnostic system in endemic areas towards strengthening the current tools and strategies for elimination of leishmaniasis. Therefore, we undertook this multicenter study for capacity development as well as clinical evaluation of the assay incorporated in the suitcase laboratory for diagnosis of VL, PKDL, and CL. 

## 2. Method and Materials

### 2.1. Study Site and Population

The study was a multi-country, single blinded, Phase 2 diagnostic evaluation study, conducted in four sites: International Centre for Diarrheal Disease Research, Bangladesh (icddr,b); Rajendra Memorial Research Institute of Medical Sciences (RMRI), India; B.P. Koirala Institute of Health Sciences (BPKIHS), Nepal, and University of Sri Jayewardenepura (USJ), Sri Lanka. For the study purpose, archived DNA samples were used to evaluate the assay. DNA isolated from clinical samples of confirmed VL, PKDL, and CL patients were used to assess the sensitivity, while samples from healthy individuals were used to determine the specificity of RPA assay. Following extraction from clinical samples, isolated DNA samples were stored at −80 °C or −20 °C in each of the study sites.

A total of 450 clinical samples from VL patients, 429 from PKDL patients, 47 from CL patients, and 322 from endemic healthy/healthy controls having been collected for research purposes by following the proper consenting procedure in earlier studies, were included in this study. Samples were archived for research purposes with due consent from the participants. During the study, all samples were handled anonymously to ensure the privacy and anonymity of the participants. 

The operational definitions for the study participants from whom the clinical samples were collected are defined below:

**VL:** An individual from a VL endemic area of either sex with no history of previous VL, suffering from fever for more than two weeks with splenomegaly and being positive for rK39 RDT. Further, the individual to respond after receiving the treatment for VL and to follow-up for cure assessment.

**PKDL:** A treated VL case of either sex from a VL endemic area presenting with skin lesions with preserved skin sensitivity and being positive for rK39 RDT. 

**CL:** An individual from a CL endemic area of either sex who had never travelled to a leishmaniasis endemic country, presented with a chronic non-healing skin lesion and positive in punch biopsy for Leishmania DNA detected by PCR.

**Endemic Healthy controls for VL and PKDL:** An individual from a VL endemic area of either sex having no history of VL/PKDL, clinically healthy without any symptoms of severe acute or chronic illness including VL and PKDL, and being rK39 RDT negative.

**Non-endemic controls for CL:** Discarded skin samples obtained from a Surgical Casualty Theatre in a non-endemic region with skin sample PCR being negative for *Leishmania* PCR.

### 2.2. Study Design

This multi-country study was conducted in two phases—(A) Preparatory Phase and (B) Implementation Phase (Figure 1). In the preparatory phase, archived clinical samples were selected and retrieved as per the study’s operational definition for each category. After that, a DNA library was prepared from the clinical samples by following the site specific DNA extraction method. In the implementation phase, capacity was developed in each of the sites to evaluate the RPA assay. At first, the mobile suitcase laboratory was deployed at each study site, and then a standardized uniform training was provided on RPA assay to the laboratory personnel in each site before starting the procedure (Figure 2). After successful completion of the training, the site investigator generated random codes for each sample to ensure the laboratory personnel were blind of the classification or type of the sample. Following de-identification, DNA samples were subjected to both real-time PCR and RPA assay as the reference method and the index method, respectively. After completion of all laboratory analyses at each site, all laboratory data were compiled, and shipped to data management center at icddr,b for analysis. For quality check, a portion of raw RPA data was checked by the designated expert prior to final data analysis. In addition, well-characterized positive and negative controls were included in each run to assure the handling and reagent quality. Besides, to ensure the quality of the study activities independent reviewers from PATH, India performed monitoring and evaluation periodically.

### 2.3. Assembly of the Mobile Suitcase Laboratory

The total set up consisted of two mobile suitcases laboratories (Figure 2C), which were needed for separate workspaces for nucleic acid extraction and detection, to avoid any possible contamination. Of them, one was designed to host all the reagents and equipment to perform the magnetic bead based nucleic acid extraction method (SpeedXtract, Qiagen, Hilden, Germany), while the other was used to perform isothermal nucleic acid amplification using a detection technology. The main idea invoked to tailor this total setup involves garnering a range of portable equipment in a water and dust resistant suitcase, which is easy to transport and store the equipment and reagents, and moreover, the suitcase can be useful to perform the test directly. Each suitcase was with case with 56 × 45.5 × 26.5 cm (extraction suitcase) and 63.1 × 50 × 30.2 cm (detection suitcase) in size (Peli case, Berlin Germany) (Figure 2). The bottom of the suitcase was stuffed with foam cubes to absorb shocks. On the top of the foam, a PVC layer was fixed that contained cutouts to host the equipment. All the edges around the equipment and the edges of the case on the PVC layer were glued with hot melt adhesive material. The mobile set up was powered by a solar panel and power pack (Yeti 400 set, GOALZERO, South Bluffdale, UT, USA) (Figure 2). The fully charged battery powers the two laboratories for up to 18 h. 

### 2.4. Laboratory Analysis

#### 2.4.1. Real-Time PCR for Detection of LD DNA

In icddr,b, the real time PCR was performed according to a method originally described by Vallur et al. [23]. Briefly, Taqman primers and probes were designed targeting the conserved region of *Leishmania* REPL repeats (L42486.1) specific for *L. donovani* and *L. infantum* and synthesized by Applied Biosystems. Briefly, a 20 μL reaction mix was prepared containing 5 μL template, 10 μL of TaqMan^®^ Gene Expression Master Mix (2X), 1 μL pre-ordered primer-probe mix, and PCR grade water. Amplification was performed on a Bio-rad CFX96 iCycler system with the following reaction conditions: 10 min at 95 °C, followed by 45 cycles of 15 s at 95 °C and 1 min at 60 °C. To quantify the parasite load of each sample, each run included one standard curve with DNA concentrations ranging from 10,000 to 0.1 parasites per reaction. Each run also included one reaction with molecular grade water as a negative control. Each DNA sample was analyzed in duplicate and in the case of an indeterminate result, one additional analysis was performed. Samples with cycle threshold (Ct) >40 were considered negative.

In USJ, real-time PCR based on the ITS-1 region of LD was conducted as described by El Tai et al. [14] using the QuantiTect SYBR Green PCR Kit (Qiagen, Hilden, Germany) and Bio Rad CFX96 Real-Time PCR Machine. The mix was prepared according to the manufacturer’s instructions and the following temperature profile was carried out: 95 °C for 15 min, then 40 cycles of 95 °C for 20 s, 53 °C for 30 s, and 72 for 1 min. For the melting curve analysis, 50 °C to 95 °C, increment 0.5 °C, for 0:05 was accomplished. A final cooling step at 25 °C for 30 s was performed. 

In BPKIHS, TaqMan primers and probe targeting the conserved region of Leishmania kDNA minicircles specific for *L. donovani* and *L. infantum* were used [16]. LightCycler^®^ 480 Probes Master (Roche, Mannheim, Germany) and Rotor gene Q system (Qiagen, Hilden, Germany). Amplification was performed with the following reaction conditions: 10 min at 95°C, followed by 45 cycles of 15 s at 95 °C, and 1 min at 60 °C.

In RMRI, real-time PCR amplifying the kDNA gene was performed according to the method described elsewhere [13] and was conducted by LightCycler^®^ 480 Probes Master (Roche, Mannheim, Germany) according to the manufacturer’s instruction. The real-time PCR protocol on the Bio Rad CFX96 Real-Time System was as follows: 5 min at 95 °C, followed by 40 cycles of 10 s at 95 °C, and 40 s at 54 °C.

#### 2.4.2. RPA Assay for Detection of LD DNA

Several types of equipment including a tube scanner (T8, Axxin, Fairfield, Australia) were used to perform the RPA assay in the detection suitcase. First, 13 μL of the RPA primers and probe mix were added to the lid of the 0.2 mL tube containing RPA lyophilized pellet (TwistAmpexo kits, TwistDx, Cambridge, UK) at concentration of 420 nM and 120 nM, respectively. Then 32 μL of rehydration buffer containing 14 mM Mg acetate followed by 5 μL of template were added. After that, the tube was closed and mixed well before it was placed into the tube scanner and incubated for 15 min at 42 °C. A mixing step was crucial after 3 min to break the goblets to facilitate the amplification step of the RPA. The emitted fluorescence signals were measured at 20 s intervals. A combined threshold and first derivative analysis was used for signal interpretation. The total time for RPA reaction, including handling was approximately 20 min. The above method has already been described in a previous study by Mondal et al. [21].

### 2.5. Data Analysis

Parametric and non-parametric tests were performed based on the distribution of data. Standard statistical formulas were followed to determine the sensitivity and specificity of the tests with 95% CI. Cohen’s kappa coefficient (k) and McNemar’s test were performed to determine the concordance and discordance between the reference and index diagnostic methods. Furthermore, receiver operating characteristic (ROC) curve analysis was performed to determine the accuracy of each of the laboratory methods. All statistical analyses were performed using SPSS (Version 25.0) and GraphPad Prism (Version 8.1.2). P value < 0.05 was considered as statistically significant.

## 3. Results

### 3.1. Performance of Real-Time PCR in Diagnosis of VL, PKDL, and CL

Real-time PCR was found to be highly sensitive in detecting leishmania DNA in clinical samples from VL patients. The observed sensitivity for real-time PCR ranged from 94–100% in different centers when DNA was extracted from blood (Table 1). Equivalent sensitivity (97%) was observed when the DNA was extracted from bone-marrow (BM). Interestingly, the pooled sensitivity of real-time PCR was also found to be around 97% (Table 1). For PKDL, the sensitivity of the assay was variable among the centers. The sensitivity of real-time PCR was found to be three times higher in detecting the parasite DNA from blood samples of PKDL patients at RMRI, than that of the pooled sensitivity (25%) (Table 2). Since parasite is abundant in skin lesions of PKDL patients, promising sensitivity of real-time PCR was noted both at icddr,b (100%) and RMRI (93%) when DNA was isolated from skin samples (Table 2). As of PKDL, real-time PCR showed excellent positivity rate in detection of LD DNA in skin samples of CL patients where all the skin samples (N = 47) were detected positive with 100% (95% CI:92.45–100.00) sensitivity. At BPKIHS and icddr,b, the performance of real-time PCR was absolute (100%) in terms of identifying true negative healthy controls, whereas markedly lower specificity (~80%) was observed at RMRI (Table 3). In ROC analysis, blood samples were found to be excellent in determining the diagnostic accuracy of real time PCR for VL. Likewise, the assay showed optimum diagnostic accuracy when skin samples were used for PKDL (Table 5) (Figure 3). At USJ, all the collected skin samples from non-endemic controls were detected negative with 100% (95% CI:69.15–100.00) specificity.

### 3.2. Performance of RPA Assay in Diagnosis of VL, PKDL, and CL

In the quest of determining the diagnostic efficiency, all the 1196 samples were subjected to RPA assay. Except for the skin samples from CL patient in USJ, a promising concordance between real-time PCR and RPA assay was observed at all sites (Table 4). For VL patients, the pooled sensitivity was found to be 88% when the blood DNA was subjected to this new isothermal assay (Table 1). Likewise, the observed sensitivity of the RPA assay ranged from 86–92% among the study sites while diagnosing the VL patients. Interestingly, identical sensitivity (97%) was found for both real-time PCR and RPA assay when BM DNA was used to perform the molecular methods (Table 1). Like real-time PCR, RPA assay showed a weak positive rate (20%) in detecting LD DNA from blood of PKDL patients, whereas three times higher sensitivity was noted at RMRI than that of the pooled sensitivity of the assay. Furthermore, the assay was found to be less sensitive at RMRI compared to icddr,b in detecting the parasite from skin samples of PKDL patients where the detected sensitivities for RPA assay were 88% and 99%, respectively (Table 2). Notwithstanding, the promising efficacy of RPA assay in diagnosing both VL and PKDL patients, the assay showed a poor sensitivity of 55.32% (95% CI: 40.12–69.83) in detecting parasite in skin samples of CL patients. However, absolute specificity was found for RPA assay in detecting controls at icddr,b and BPKIHS. Moreover, this assay was found to be 100% (95% CI: 69.15–100.00) specific in detecting non-CL participants at USJ. In contrast, a suboptimal specificity (79%) was observed for RPA assay at RMRI in detecting true negatives (Table 3). Finally, the diagnostic accuracy of RPA assay was found to be comparable to real-time PCR while considering the area under the curve (AUC) determined through ROC analysis (Table 5) (Figure 3).

## 4. Discussion

The overarching goal of our study was to evaluate the feasibility, applicability, and efficacy of RPA assay which has a multitude of advantages over the existing isothermal amplification technologies for molecular diagnostics.

With the auspices of multifarious advantages of RPA assay incorporated in the suitcase laboratory, we undertook this multi-center blind study for establishment of this point-of-need diagnostic system towards strengthening the current diagnostic platforms for detection of leishmaniasis in endemic settings. Following successful capacity development at each study site, a total of 1196 clinical samples was subjected to clinical evaluation through both index and reference methods. We observed promising diagnostic efficiency of RPA assay compared to real-time PCR in detection of both VL and PKDL, where the method was found to be user-friendly to the laboratory personnel. Surprisingly, the method showed poor sensitivity for CL, whereas the specificity was found to be equivalent to that of the reference molecular method. Following laboratory analysis of all clinical samples, we found excellent agreement between the molecular methods; however, at USJ the agreement was poor (Table 4).

A marginal difference in sensitivity was observed between RPA assay and real-time PCR at each site in the diagnosis of VL from blood derived DNA, so that the pooled sensitivity RPA assay was less compared to the pooled sensitivity of real-time PCR. The relatively higher sensitivity of real-time PCR at each site, especially at RMRI can be attributed to the improved analytical sensitivity of real-time PCR methods performed at each site [14,16,23]. Moreover, the burden of parasite in the clinical samples was different. Ultimately, samples with low parasite burden or higher Ct value in real-time PCR were more likely to be negative in RPA assay. Furthermore, RPA assay was performed following real-time PCR to detect the LD DNA which might have ensued with degradation of LD DNA due to several cycles of freeze thawing. A recent publication on the diagnosis of SARS-CoV-2 through RPA assay stated the importance of using fresh samples to validate new diagnostics [24]. In addition, the archived clinical samples for diagnosis of VL used in this study were not homogeneous among the study sites. Only blood samples were collected at both icddr,b and RMRI whereas bone marrow samples were archived as well at BPKHIS. Earlier studies performed on both blood and bone marrow samples with real-time PCR methods that were followed in this study showed equivalent diagnostic efficiency as well [13,15,16,25]. Interestingly, unlike blood samples, equal sensitivity was observed for RPA and real-time PCR when BM derived DNA was used which should be attributable to a higher parasite burden in BM.

Despite a downward trend of VL in the Indian subcontinent, the PKDL cases are spiking in Bangladesh, India, and Nepal. To date, early diagnosis and treatment has been the mainstay in curbing these biological engines towards restricting inter-epidemic disease transmission [26]. In the current study, both blood and skin samples were collected to investigate the presence of LD DNA in PKDL patients having skin lesions/rash. We found a comparable sensitivity of RPA assay to real-time PCR in the diagnosis of PKDL cases through skin samples. As expected, a very poor positive rate was observed for blood samples, whereas more than half of the PKDL cases at RMRI were found to be positive through both RPA assay and real-time PCR. A previous study performed by Mondal et al. reported around 50% sensitivity of PCR towards diagnosis of suspected PKDL cases [5]. On the other hand, a number of studies reported very poor sensitivity of NAAT assays in the diagnosis of PKDL cases with blood samples [11,17,27]. It is assumed that for PKDL cases parasite resides in the skin mostly, whereas the blood contains scanty parasites. Here of note, the recent xenodiagnoses study performed by Mondal et al. showed high infectiousness of PKDL cases towards sterile sand flies which confers the presence of parasite in peripheral blood streams [28]. In addition to the variability in clinical samples, the disparate sensitivities are attributable to the different forms of PKDL cases [11,29,30].

In this study, skin derived DNA from CL cases was subjected to both RPA assay and real-time PCR at USJ in Sri Lanka. Since the parasite resides in the skin for CL cases, therefore, only dermal samples were used for diagnosis purposes. Surprisingly, the sensitivity for real-time PCR was almost two times higher than that of RPA assay towards diagnosis of CL cases. The varied sensitivity could be attributed to the sequence variability of CL causing LD, while the RPA assay was designed based on the sequence of VL causing LD parasite. Surprisingly, the previous study showed promising efficiency of the RPA method for diagnosis of CL, however, the sensitivity of real-time PCR remains similar to the previous findings [14,22]. The real-time PCR for the detection of the LD from CL, patient targeted the ITS genome region, while the RPA is based on the kDNA gene. The latter showed a limited sensitivity for CL patients even for real-time PCR [31].

Notwithstanding, the promising sensitivity of both molecular methods, limited specificity was observed in India, whereas absolute specificity was found for both of the methods in icddr,b and BPKHIS. The previous laboratory-based study performed by Mondal et al. corroborates the findings of the current study [21]. However, the paradox at RMRI might be attributed to the high residual infection or asymptomatic infection among the apparently healthy individuals in India [24,32,33]. On the contrary, in Bangladesh and Nepal, the disease burden has been very low for the last few years, eventually, the likelihood of asymptomatic infection is low. Moreover, rK39 RDT negatives were included as endemic healthy controls in the current study, whereas few studies reported poor agreement between serological tests and molecular tests towards detection of asymptomatic infection [31,34]. In Sri Lanka, 100% specificity was observed for both RPA assay and real-time PCR while diagnosing the skin samples collected from endemic healthy controls.

With no exception, the study findings were compromised due to several biological and technical drawbacks. First, the collection and preservation procedure were not similar at each site for the clinical samples being used in this study. Moreover, the downstream methods including DNA extraction and real-time PCR methods followed in each site were also different. Apart from the technical aspects, the efficacy of the index method varied due to the different sets of participants enrolled at each site. Notably, the severity of the diseased participants was different at each site, which has a correlation to the parasite burden in the clinical samples [11,15,35]. Thus, the performance of the index assay is likely to have been overestimated if there was disproportionate inclusion of patients having high parasite burden or degree of severity and vice versa. Such limitations reinforce the need of a standardized molecular method for LD diagnostics. Finally, blood samples from endemic controls were used to determine the specificity of the assays, whereas skin samples were collected from PKDL patients to determine the sensitivity. However, previous studies substantiated the specificity of the assays through using blood samples from non-VL participants [11,30].

This very first multi-center study has addressed crucial factors including the patient type, a suitable clinical sample, and the technical capacity of different endemic settings towards evaluating the application of RPA assay incorporated in a mobile suitcase laboratory. Considering the versatile advantages along with this study’s findings, the LD RPA assay was proved to be promising in accordance to the target product profile (TPP) for point-of-need diagnosis of Chagas disease and dermal leishmaniasis [36,37]. Furthermore, this assay has the potential to meet the guidelines outlined by the World Health Organization (WHO) that diagnostics for developing countries should be ASSURED: Affordable, Sensitive, Specific, User-friendly, Rapid and robust, Equipment-free, and Deliverable to end users [38]. However, the DNA extraction method is still challenging to perform downstream molecular analysis at point-of-care. Recently, a study by Chowdhury et al., showed the promises of a rapid extraction method for isolation of DNA from skin samples [30]. Earlier, we showed the excellent efficiency of this method in diagnosing VL from blood samples at a field site [21]. Findings of the current study will facilitate future studies with similar clinical samples collected from properly categorized leishmaniasis patients with similar disease patterns where harmonization of DNA extraction methods must be ensured. This study is an exemplary study towards establishment of molecular methods in different endemic settings. The current RPA assay is based only on the minicircle DNA sequence from *L. donovani* which can detect, *aethiopica, infantum* and *major* [21]. However, the development of a pan-RPA assay is prerequisite towards application of this rapid diagnostic tool both in the old and new worlds. Recently, further improvement of this assay was done towards detection of CL. RPA assay targeting the 18S gene has shown a good sensitivity for samples collected from CL (personal communication). Further improvisation has been done to this method for visualization of the reaction outcome in RDT or fluidics formats [18,39]. Apart from diagnosis, RPA assay might be useful for prognosis, as real-time PCR is still considered to be the most effective tool for prognosis [17]. Furthermore, this method would also be potentially useful for vector surveillance or molecular xeno-monitoring studies. Another use of this method is that RPA assay products are viable for sequencing studies which might facilitate the investigation of potential drug resistance in clinical isolates from leishmaniasis patients [18,40]. Finally, a future large-scale prospective study should be performed to discern the proper utility of this rapid molecular assay at patient level. Besides, the mobile laboratory set-up would be a potential ancillary in emergency response to combat infectious disease outbreaks.

## Figures and Tables

**Figure 1 microorganisms-09-00588-f001:**
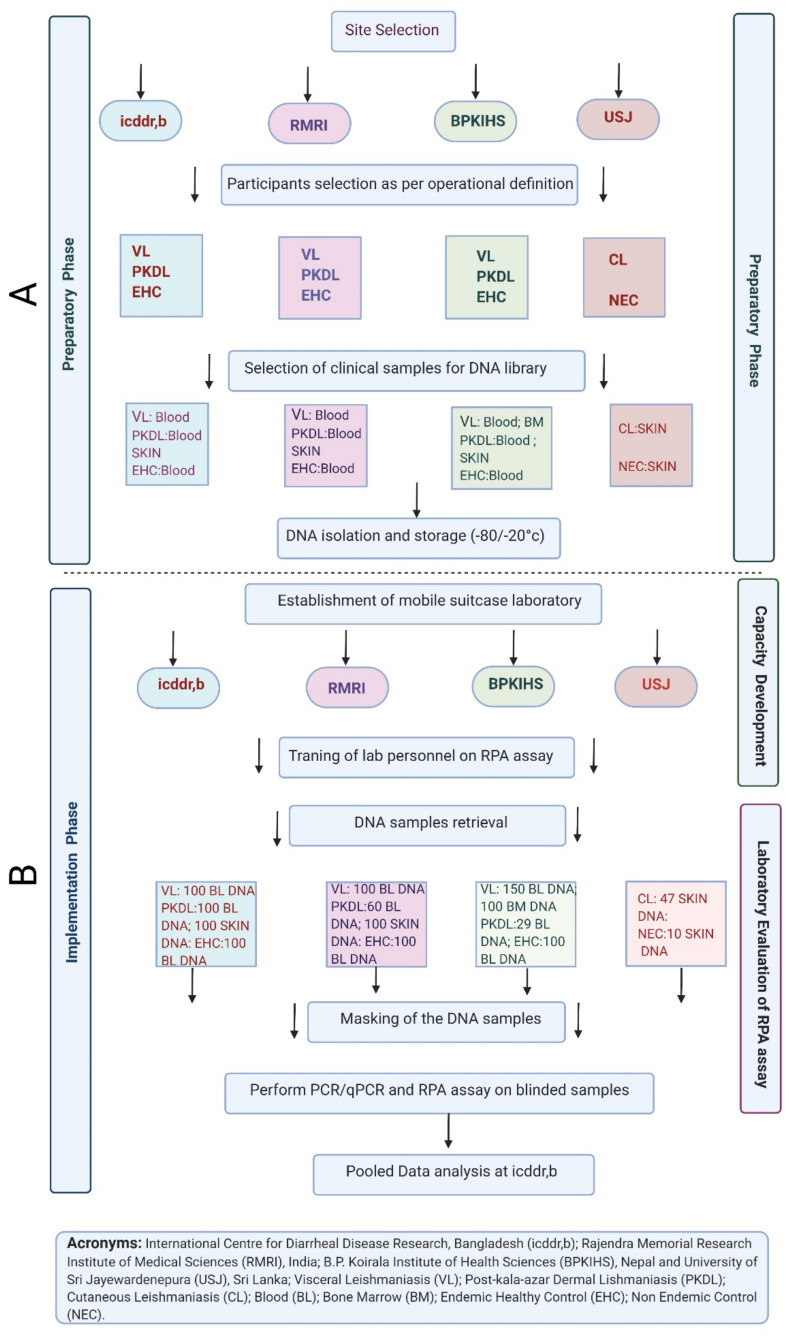
The schematic representation of two phases—(**A**) Preparatory Phase and (**B**) Implementation Phase, towards multi-site evaluation of LD RPA assay incorporated in the suitcase laboratory.

**Figure 2 microorganisms-09-00588-f002:**
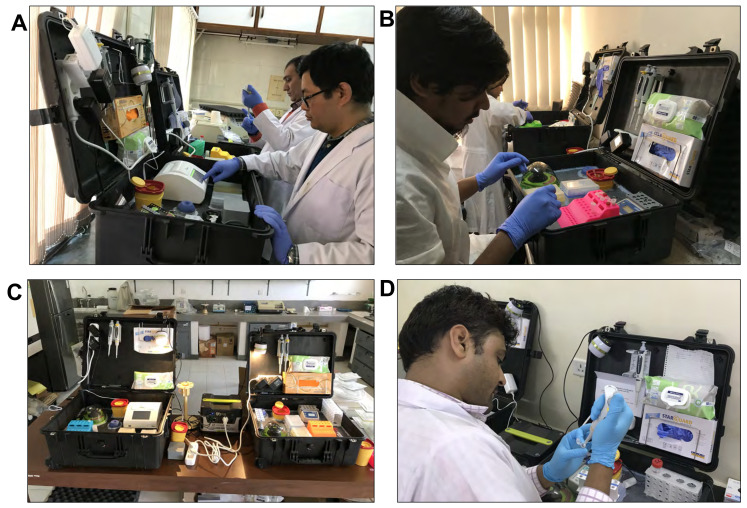
Deployment of the mobile suitcase lab in (**A**): B.P. Koirala Institute of Health Sciences (BPKIHS), Nepal; (**B**): International Centre for Diarrheal Disease Research, Bangladesh (icddr,b); (**C**): University of Sri Jayewardenepura (USJ), Sri Lanka and (**D**): Rajendra Memorial Research Institute of Medical Sciences (RMRI), India.

**Figure 3 microorganisms-09-00588-f003:**
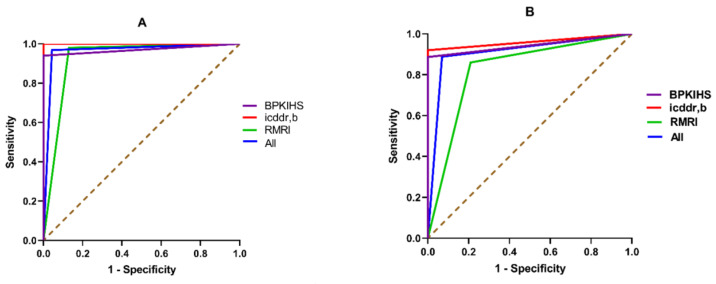
The receiver operating characteristic (ROC) curves depict the diagnostic accuracy of real-time PCR and RPA assays in diagnosis of both VL and PKDL with blood and skin specimens, respectively. (**A**) Area under the curves for real-time PCR in diagnosis of VL; (**B**) Area under the curves for RPA in diagnosis of VL; (**C**) Area under the curves for real-time PCR in diagnosis of PKDL; (**D**) Area under the curves for RPA in diagnosis of PKDL.

**Table 1 microorganisms-09-00588-t001:** Clinical sensitivity of both real-time PCR and LD RPA assay using blood and bone marrow samples of VL patient from three centers in Bangladesh, India, Nepal.

Site	Sample Type	Sensitivity (% CI) (n/N) Real-Time PCR	Sensitivity (% CI) (n/N) RPA	Pooled Sensitivity (% CI) (n/N) Real-Time PCR	Pooled Sensitivity (% CI) (n/N) RPA
icddr,b	Blood DNA	100% (96.38–100) (100/100)	92% (84.84–96.48) (92/100)	96.86% (94.45–98.42) (339/350)	88.85% (85.08–91.96) (311/350)
RMRI	Blood DNA	98% (92.96–99.76) (98/100)	86% (77.63–92.13) (86/100)		
BPKIHS	Blood DNA	94% (88.92–97.22) (141/150)	88.7% (82.48–93.26) (133/150)		
BPKIHS	bone marrow DNA	97% (91.48–99.38) (97/100)	97% (91.48–99.38) (97/100)	NA	NA

**Table 2 microorganisms-09-00588-t002:** Clinical sensitivity of both real-time PCR and LD RPA assay using blood and skin samples of PKDL patient from three centers in Bangladesh, India, Nepal.

Site	Sample Type	Sensitivity (% CI) (n/N) Real-Time PCR	Sensitivity (% CI) (n/N) RPA	Pooled Sensitivity (% CI) (n/N) Real-Time PCR	Pooled Sensitivity (% CI) (n/N) RPA
icddr,b	Blood DNA	1% (0.03–5.45) (1/100)	0.0% (0.00–3.62) (0/100)	24.33% (18.40–31.10) (46/189)	20.10% (14.64–26.54) (38/189)
RMRI	Blood DNA	75% (62.14–85.28) (45/60)	63.3% (49.90–75.41) (38/60)		
BPKIHS	Blood DNA	0.0% (0.00–11.94) (0/29)	0.0% (0.00–11.94) (0/29)		
icddr,b	Skin DNA	100% (96.38–100) (100/100)	99% (94.55–99.97) (99/100)	96.50% (92.92–98.58) (193/200)	93.50% (89.14–96.49) (187/200)
RMRI	Skin DNA	93% (86.11–97.14) (93/100)	88% (79.98–93.64) (88/100)		

**Table 3 microorganisms-09-00588-t003:** Specificity determination of both real-time PCR and LD RPA assay in diagnosis of VL and PKDL.

Site	Sample Type	Specificity (% CI) (n/N) Real-Time PCR	Specificity (% CI) (n/N) RPA	Pooled Specificity (% CI) (n/N) Real-Time PCR	Pooled Specificity (% CI) (n/N) RPA
icddr,b	Blood DNA	100.00% (96.38–100.00) 100/100	100.00% (96.38–100.00) (100/100)	96% (93.12% to 97.92%) (288/300)	93.00% (89.50% to 95.61%) (279/300)
RMRI	Blood DNA	88.00% (79.98–93.64) (88/100)	79.00% (69.71–86.51) (79/100)		
BPKIHS	Blood DNA	100.00% (96.34–100.00) (100/100)	100.00% (96.34–100.00) (100/100)		

**Table 4 microorganisms-09-00588-t004:** Agreement between real-time PCR and RPA assay at different sites.

Site	Concordance (Real-Time PCR vs RPA); Kappa Value	Agreement	*p* Value
icddr,b	0.950	Excellent	0.016
RMRI	0.781	Good	0.035
BPKIHS	0.900	Excellent	0.023
USJ	0.303	Weak	0.087
All sites	0.855	Excellent	0.015

**Table 5 microorganisms-09-00588-t005:** Determination of the area under the curve (AUC) for each of the methods in diagnosis of both VL and PKDL with blood and skin specimens, respectively.

Disease Type	Site	Sample Type	Area Under the Curve (95% CI)	
			Real-Time PCR	RPA
Visceral Leishmaniasis	BPKIHS	Blood DNA	0.97 (0.95–0.99)	0.94 (0.91–0.97)
	icddr,b		1.00 (1.00–1.00)	0.96 (0.93–0.99)
	RMRI		0.93 (0.88–0.97)	0.83 (0.76–0.89)
	All		0.96 (0.95–0.98)	0.91 (0.88–0.94)
PKDL	icddr,b	Skin DNA	1.00 (1.00–1.00)	0.96 (0.98–1.00)
	RMRI		0.91 (0.86–0.95)	0.84 (0.76–0.89)
	All		0.95 (0.93–0.98)	0.92 (0.88–0.95)

## Data Availability

The data presented in this study are available on request from the corresponding authors. The data are not publicly available due to variable institutional data policies at each study site.

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
