# Peer review of "A Multi-Country, Single-Blinded, Phase 2 Study to Evaluate a Point-of-Need System for Rapid Detection of Leishmaniasis and Its Implementation in Endemic Settings"

_microorganisms, 2021, doi:10.3390/microorganisms9030588_

Round 1

Reviewer 1 Report

The manuscript “A multi-country, single-blinded, phase 2 study to evaluate a point-of-need system for rapid detection of leishmaniasis and its implementation in endemic settings” describes the efficacy of RPA assay which has multitude of advantages over the existing isothermal amplification technologies for molecular diagnostics. This paper has important clinical message and should be of great interest to the readers. However it is advisable to reformulate the general structure of the paper to focus the results obtained and it should be made clearer why there are differences in sensitivity in India.

Author Response

The manuscript “A multi-country, single-blinded, phase 2 study to evaluate a point-of-need system for rapid detection of leishmaniasis and its implementation in endemic settings” describes the efficacy of RPA assay which has multitude of advantages over the existing isothermal amplification technologies for molecular diagnostics. This paper has important clinical message and should be of great interest to the readers. However, it is advisable to reformulate the general structure of the paper to focus the results obtained and it should be made clearer why there are differences in sensitivity in India.

Response: We appreciate your judicious comment towards improving the manuscript with a vivid projection on the findings. The discussion section has been revised to be more focused on the results. The plausible explanation on the difference in sensitivity between the molecular methods has been outlined in the revised discussion (Line:342-351). Furthermore, we addressed the issue as a limitation as well (Line:400-408).

Reviewer 2 Report

In the manuscript A multi-country, single-blinded, phase 2 study to evaluate a point-of-need system for rapid detection of leishmaniasis and its implementation in endemic settings, the authors evaluate two molecular diagnostic methods (qPCR and RPA) for the detection of leishmaniasis. Sensitivity and specificity are good. However, something that caught my attention is that the research centers use different amplification equipment and different PCR conditions.

It would be interesting to demonstrate whether the differences in sensitivity are the result of the experimental procedure, for this, perhaps they can show us the results of tests with double-blind samples, where controls are included that allow us to see the detection limit that each equipment presents with the PCR conditions used by each research center.

Author Response

In the manuscript A multi-country, single-blinded, phase 2 study to evaluate a point-of-need system for rapid detection of leishmaniasis and its implementation in endemic settings, the authors evaluate two molecular diagnostic methods (qPCR and RPA) for the detection of leishmaniasis. Sensitivity and specificity are good. However, something that caught my attention is that the research centers use different amplification equipment and different PCR conditions.It would be interesting to demonstrate whether the differences in sensitivity are the result of the experimental procedure, for this, perhaps they can show us the results of tests with double-blind samples, where controls are included that allow us to see the detection limit that each equipment presents with the PCR conditions used by each research center.

Response: Thank you so much for reviewing the manuscript. We appreciate your meticulous observations. Indeed, different amplification equipment and different PCR conditions were followed at different study sites. Moreover, the panels of clinical samples were different as well. Eventually, it is evident that multiple factors are responsible for variable performance of RPA assay at different sites. The main goal of this study was to develop a unified molecular method for detection of parasite in clinical samples to minimize the technical caveats. However, the type of clinical samples will remain the critical player in addressing such deviations in performance of the method. We have addressed the issues in discussion section (Line: 400-408) in the light of empirical evidence.  

Reviewer 3 Report

Dear Authors,

You address a major problem. Rapid, relatively simple conformation of a Leishmaniasis diagnosis in an endemic peripheral setting. I read it with interest, because for me was the solution original and new. The suitcase laboratory was not new but in this sophisticated way for me yes. The new isotherm RPA assay gave the possibility. You did this study to evaluate its diagnostic properties still in sophisticate centres and found it very suitable surely for kala-azar and PKDL, where it is most necessary. I wait anxious for a real field trial. Hope that for other species than L. donovani it also will be available. The paper is exceedingly long and could be more concise. Particularly the discussion. In the discussion you go much further than discussion of the new equipment. It is nice but I am afraid that the focus is lost.

I will go through the paper to give a few comments:

In study site and population, I wondered whether the social standard of the controls will be different from the patients.

I noticed the equipment and expendables are not that cheap compared with the cost of the old established methods, looking at obtaining the material and searching for LD bodies. In a time of decreasing knowledge and education an expensive laboratory test may be the answer but will further decrease experience and knowledge. This will not diminish the appreciation of the equipment and thoughts.

You carefully assess PCR in the different centres in different material, Blood, bone marrow and skin, sensitivity, and specificity.

Interesting is the observation that both tests PCR and RPA find lower positives in PKDL. Since most of the parasites are in the skin it is most important how and where the blood is taken. How much of tissue fluid is also ingested. But the skin itself is most important, as you already concluded.

The sandfly does not sting directly in the bloodvessel but passed through the tissues. That is how it becomes infected, I think.

Why not just compare traditional PCR with the RPA. Then see that it is not always comparable and explain. Explain the difference of the results in the different test sites and why. But the way you do it now is may be better, but difficult for clinicians. May be not for laboratory people but I also think they consider it to go further then showing that it works. It seems you are most interested in the test as an epidemiological tool, although at the end you state it should also be assessed at patient level.. Is it needed to compare with the other tests as well, when you did not do them on your samples?

Author Response

You address a major problem. Rapid, relatively simple conformation of a Leishmaniasis diagnosis in an endemic peripheral setting. I read it with interest, because for me was the solution original and new. The suitcase laboratory was not new but in this sophisticated way for me yes. The new isotherm RPA assay gave the possibility. You did this study to evaluate its diagnostic properties still in sophisticate centres and found it very suitable surely for kala-azar and PKDL, where it is most necessary. I wait anxious for a real field trial. Hope that for other species than L. donovani it also will be available. The paper is exceedingly long and could be more concise. Particularly the discussion. In the discussion you go much further than discussion of the new equipment. It is nice but I am afraid that the focus is lost.

Response: We appreciate your insightful comments on the finding of our study. Indeed, the promises of RPA assay incorporated in suitcase laboratory for detection of infectious diseases have already been proven. The current study will accentuate the implementation of the mobile set up at remote settings. In our previous study demonstrated the feasibility of the suitcase lab at field condition for diagnosis of VL [21]. Considering the efficiency and feasibility of the rapid assay, it is not far away for implementation at field setting for detection of leishmania infection in endemic tropics. Now, it is necessary to develop pan-leishmania RPA assay for detection of the parasite at both old and new worlds.

The discussion section is more vivid now with a concise impression on the findings.

I will go through the paper to give a few comments:

In study site and population, I wondered whether the social standard of the controls will be different from the patients.

Response: The controls were from the similar endemic areas of their corresponding cases. Therefore, it is more likely that both cases and controls share similar social standard.

I noticed the equipment and expendables are not that cheap compared with the cost of the old established methods, looking at obtaining the material and searching for LD bodies. In a time of decreasing knowledge and education an expensive laboratory test may be the answer but will further decrease experience and knowledge. This will not diminish the appreciation of the equipment and thoughts.

Response: Indeed, the test is not as cheap as the rapid diagnostic tests. However, we found the test is three times cheaper than real-time PCR in our previous study [29]. Furthermore, the method is less complicated than PCR based molecular methods. 

You carefully assess PCR in the different centres in different material, Blood, bone marrow and skin, sensitivity, and specificity.

Response: We tried to determine the efficiency of the method for detection of LD DNA in variable clinical samples.

Interesting is the observation that both tests PCR and RPA find lower positives in PKDL. Since most of the parasites are in the skin it is most important how and where the blood is taken. How much of tissue fluid is also ingested. But the skin itself is most important, as you already concluded.The sandfly does not sting directly in the bloodvessel but passed through the tissues. That is how it becomes infected, I think.

Response: Your assumption is rational. We will be happy if you kindly go through our articles on xeno-diagnosis to understand the transmission of parasites from reservoir to vector [27].

Why not just compare traditional PCR with the RPA. Then see that it is not always comparable and explain. Explain the difference of the results in the different test sites and why. But the way you do it now is may be better, but difficult for clinicians. May be not for laboratory people but I also think they consider it to go further then showing that it works. It seems you are most interested in the test as an epidemiological tool, although at the end you state it should also be assessed at patient level.. Is it needed to compare with the other tests as well, when you did not do them on your samples?

Response: The traditional PCR method is less sensitive and time consuming, therefore, it is not being performed for detection of the parasite. In our previous study, we showed multiple advantages of real-time PCR towards diagnosis and treatment monitoring of both VL and PKDL patients [15]. Since, the RPA assay gives the result in real time, therefore, it is way more rational to compare it to another real-time assay.

Currently, real-time PCR is being used by the clinicians for investigating the patients. RPA assay is far more simple method compared to PCR based methods. Eventually, the application of RPA assay for making clinical decisions will be more useful and pragmatic.

Round 2

Reviewer 1 Report

The paper is accepted without any further changes.